# Association between cardiometabolic risk factors and multidrug-resistant tuberculosis: A case-control study

Sishir Poudel[1,2]*, Laxman Wagle[3], Tara Prasad Aryal[4], Binay Adhikari[2☼], Sushan Pokharel[1☼], Dipendra Adhikari[2☼], Kshitiz Bhandari[1☼], Kshitiz Rijal[1☼], Jyoti Bastola Paudel[5☼]

1 BP Koirala Institute of Health Sciences, Dharan, Province 1, Nepal, 2 Tuberculosis Treatment Center, Pokhara, Gandaki Province, Nepal, 3 Department of Internal Medicine, Ascension Saint Agnes Hospital, Baltimore, Maryland, United States of America, 4 Ministry of Health and Population, Kathmandu, Bagmati Province, Nepal, 5 Nepal Police Hospital, Kathmandu, Bagmati Province, Nepal

☼ These authors contributed equally to this work.
* sishir.poudel.sh@gmail.com

## Abstract

### Background

Multidrug-resistant tuberculosis (MDR-TB) continues to be a major public health concern, especially in high-burden countries like Nepal. While individual risk factors are known, the cumulative impact of cardiometabolic factors on MDR-TB is not well understood.

### Methods

A health-facility-based, age- and sex-matched 1:2 case-control study was conducted at MDR-TB treatment centers in Gandaki Province, Nepal. MDR-TB patients (cases) and drug-sensitive tuberculosis (DS-TB) patients (controls) were enrolled. Cases were defined as adults (≥18 years) with confirmed MDR-TB; controls were adults with sputum-positive DS-TB. Data on sociodemographics, cardiometabolic risk factors (alcohol, tobacco, abnormal body mass index, hypertension, diabetes), TB literacy, and treatment history were collected using a structured, pretested questionnaire by trained medical officers. Data were analyzed using Stata v13.0. Binary logistic regression was used to assess associations between risk factors and MDR-TB. Ethical approval was obtained from the Nepal Health Research Council and written informed consent was obtained from all participants.

### Results

A total of 183 participants (61 cases, 122 controls) were included. Mean age of participants was 42.5 years (SD = 18.5); 73.8% were male. Most participants were from urban areas (74.9%), and 66.7% were unemployed. Cardiometabolic risk factors

**Data availability statement:** All data files are available from the figshare database. Link:http://dx.doi.org/10.6084/m9.figshare.28931690.v1.

**Funding:** The author(s) received no specific funding for this work.

**Competing interests:** The authors have declared that no competing interests exist.

were present in 79.2% of participants. Alcohol and tobacco use were reported by 59.6% and 45.9%, respectively; 9.8% had diabetes and 7.1% had hypertension. Known TB contact and prior TB history were reported by 26.8% and 31.1% respectively. In multivariate analysis, unemployment (AOR: 5.24, 95% CI: 1.33–20.64), and known TB contact (AOR: 8.89, 95% CI: 2.46–32.15) were significantly associated with MDR-TB. Cardiometabolic risk factors were not significantly associated.

## Conclusion

Known TB contact and unemployment were significantly associated with MDR-TB, while the cumulative effect of cardiometabolic risk factors showed no significant impact, indicating that interventions should prioritize established TB-related risk factors.

## Introduction

Antimicrobial resistance (AMR) poses a critical challenge to global health, undermining decades of progress in the treatment and control of infectious diseases [1]. Among these, tuberculosis (TB) remains a leading contributor to the global burden of AMR-related mortality, accounting for approximately 13% of all AMR-attributed deaths worldwide [2]. The emergence of multidrug-resistant TB (MDR-TB)—defined as resistance to at least isoniazid and rifampicin—represents a formidable threat to global TB control efforts [1]. MDR-TB is associated with prolonged treatment duration, increased healthcare costs, greater morbidity and mortality, and significantly lower treatment success rates compared to drug-sensitive TB (DS-TB) [3].

The burden of MDR-TB is especially pressing in high TB-burden countries like Nepal, which continues to report a substantial number of MDR-TB cases despite improvements in diagnostic and therapeutic strategies [4]. The spread of MDR-TB is driven by both the emergence of resistance during treatment and the transmission of resistant strains, compounded by delays in diagnosis and inadequate access to effective second-line therapies [2].

In Nepal, of estimated 1500 MDR cases per year, only 350–450 cases are reported [5]. In 2022, around 13% of people with TB had MDR-TB, making Nepal among the top 30 MDR-burden countries worldwide [4,5].

Several individual-level risk factors for MDR-TB have been identified in the literature, including previous TB treatment, poor adherence to therapy, HIV co-infection, substance use (e.g., alcohol and tobacco), and social determinants such as low income, unemployment, and poor living conditions [6].

Among the metabolic risk factors, diabetes mellitus (DM) has received considerable attention. A recent meta-analysis reported DM to be an independent risk factor for MDR-TB, particularly in cases of primary resistance [7].

Beyond diabetes, other cardiometabolic factors, including hypertension, obesity or undernutrition (abnormal BMI), alcohol use, and smoking, have been implicated in TB pathogenesis and treatment outcomes [6]. While alcohol abuse and smoking have

well-established roles in increasing susceptibility to active TB [6], their direct associations with MDR-TB remain less clear and may be mediated by factors such as poor adherence and treatment interruption [8,9]. Similarly, malnutrition has long been recognized as a contributor to immune dysfunction, increasing the risk of TB recurrence and potentially promoting treatment failure and resistance development [10].

Conversely, obesity and metabolic syndrome are associated with chronic low-grade inflammation and immune dysregulation [11]. Dysregulated cytokine production, insulin resistance, and altered macrophage and T-cell responses in obesity may influence TB pathogenesis and possibly MDR-TB risk [11]. Hypertension, while less commonly studied in the context of TB, shares pathophysiological pathways with other metabolic disorders and may contribute indirectly to immune impairment [12].

Despite these individual associations, limited research has examined the cumulative impact of multiple cardiometabolic risk factors on MDR-TB. Given the biological plausibility and increasing prevalence of NCDs in TB-endemic regions, there is a pressing need to clarify whether cardiometabolic risk profiles contribute to MDR-TB acquisition or poor treatment outcomes. Moreover, understanding these associations could inform integrated TB–NCD care models, risk stratification strategies, and public health interventions in high-burden settings like Nepal.

This study aimed to investigate the relationship between cumulative cardiometabolic risk factors; diabetes, hypertension, alcohol use, tobacco use, and abnormal BMI, and MDR-TB status, while also assessing the role of sociodemographic variables such as employment status and TB contact history. By exploring both metabolic and social determinants, we seek to contribute to the evolving understanding of MDR-TB epidemiology and inform tailored prevention strategies.

## Materials and methods

### Study design and setting

A health-facility-based, age-sex matched 1:2 case-control study was conducted from 15th September 2024–15th January 2025 at the DR-TB treatment centers in Gandaki Province, Nepal. These centers included the Tuberculosis Treatment Center (Pokhara), Madhyabindu District Hospital (Nawalpur), and Dhawalagiri Hospital (Baglung). These centers are specialized facilities providing comprehensive TB care and treatment services, including for drug-resistant tuberculosis (DR-TB). They serve diverse populations from urban, semi-urban, and rural areas, ensuring a representative sample of the region's TB patients. The choice of these centers allowed for a broad geographical representation and large number of DR-TB patients [13].

### Cases and controls selection

This case-control study recruited TB patients aged 18 years or older presenting to three designated TB treatment centers in Gandaki Province. Cases were defined as patients with new MDR-TB (without history of MDR-TB), while the controls were patients with drug-susceptible TB and both cases and controls were undergoing respective treatment for at least two weeks before the advent of data collection. This was done to remove any confounding factors related to prior treatment or disease progression. The research team identified these patients from the patient registry of the three TB treatment centers. In one of the three treatment centers, the number of eligible controls was limited, posing a challenge to achieving an adequate sample size for the control group. To address this, the research team implemented a flexible yet systematic approach by selecting additional controls from the other two treatment centers adhering to the study's inclusion criteria to maintain comparability.

A hierarchical procedure was used to match controls with cases using individual matching for sex and nearest matching for age within 5 years.

### Measurements

The primary exposure variables were five key cardiometabolic risk factors extracted from the literature:

1. Alcohol use – It was defined as current or past habitual consumption of alcoholic beverages.

2. Tobacco use – It was defined as current or past smoking or smokeless tobacco consumption.

3. Body Mass Index (BMI) – It was calculated from measured height and weight and categorized according to WHO standards: underweight (<18.5 kg/m²), normal (18.5–24.9 kg/m²), overweight (25.0–29.9 kg/m²), and obese (≥30 kg/m²). Underweight and overweight/obese were grouped as abnormal BMI.

4. Hypertension – It was based on prior diagnosis, current use of antihypertensive medications, or blood pressure ≥130/80 mmHg. Blood pressure was measured using a calibrated digital sphygmomanometer with the participant seated after resting for five minutes. The threshold for hypertension was based 2017 ACC/AHA criteria (Stage I and II hypertension) [14].

5. Diabetes Mellitus (DM) –It was identified by self-reported diagnosis, anti-diabetic treatment, or fasting blood glucose ≥126 mg/dL (Diabetes according to American Diabetes Association Criteria) [15].

Participants were further grouped into two categories based on the presence of these risk factors to assess the cumulative cardiometabolic risk in tuberculosis patient: 1. At least one cardiometabolic risk factor 2. No cardiometabolic risk factors. Additional covariates were grouped as follows:

• Socio-demographic Characteristics: Age, sex, ethnicity, place of residence (rural/urban), occupation, educational attainment, marital status, type of settlement (temporary/permanent), and household size (number of family members).

• TB Literacy: Participants' knowledge regarding tuberculosis was evaluated using a structured questionnaire covering TB transmission, prevention methods, and treatment practices.

• Disease-Specific Factors: These included history of contact with a known TB case, prior TB diagnosis (based on information from TB registry), number of TB episodes, history of TB treatment interruption (including interruptions ≥1 week), medication history, treatment outcomes from previous episodes (e.g., cured, failed), and presence of coexisting conditions such as HIV/AIDS.

## Sample size determination and recruitment

For this study, the sample size was calculated using Epi-Info 7.2 statistical software. The parameters included in the calculation were: prevalence of TB cases (controls) previously treated of 10.4% [16], odds ratio of MDR-TB for previously treated cases of 2.8 [17], a 1:2 ratio between cases and controls, 90% power and 95% confidence level.

To adjust for a non-response rate, the initial sample size of 146 was increased by 10%, resulting in a total sample size of 162, comprising 54 cases and 108 controls for the study. All cases and controls fulfilling the inclusion criteria were consecutively included in the study until the sample size was achieved.

## Data collection tools and study procedures

Data was collected face-to-face using a structured, bilingual (English and Nepali), and pre-tested survey questionnaire prepared from existing related studies. Patients were recruited over a 16-week period, from 15th September 2024–15th January 2025. After obtaining written informed consent, patients were interviewed using a standard questionnaire in English or Nepali, depending on their preference. To minimize recall bias, a Nepali calendar was used as a reference for recalling dates. Data was collected via the KoboCollect app. All interviews were conducted by properly trained medical officers.

## Data analysis

Initially, the raw data was entered into Microsoft Excel, where thorough cleaning and consistency checks were performed. Descriptive statistics were presented as mean and standard deviation or frequency and percentage. For

bivariate comparisons between MDR-TB cases and drug-susceptible TB controls, the chi-square test or Fisher's exact test was used. The dependent variable was MDR-TB status (case = 1, control = 0). The independent variables included the sociodemographic, cumulative cardiometabolic factors, literacy, and disease-specific variables described above.

Unconditional and conditional multivariate logistic regression model were used to estimate odds ratio and the associated 95% confidence intervals (CIs). Final multivariate models were created through stepwise elimination of variables of interest from bivariate analysis (p-value ≤0.25). A p-value of less than 0.05 was considered to indicate statistical significance. Stata 13.0 software was used for all analyses.

### Ethical considerations

Ethical approval was taken from Nepal Health Research Council (Protocol Registration No: 441_2024). Permission was also taken from the hospital administration to approach participants for data collection. All participants provided written informed consent prior to inclusion in the study. Although the study did not involve any clinical procedures or interventions, participants were informed about the study's purpose, confidentiality, and voluntary nature of participation. Only adults (aged 18 years and above) were included, so no parental or guardian consent was required. Anonymity of participants was assured in the entirety of the study.

## Results

A total of 61 cases and 122 matched controls were interviewed for the study. Table 1 demonstrates the background characteristics of cases and controls.

A total of 183 participants were included in the study, with a mean age of 42.52 ± 18.46 years. The age distribution suggests most participants in adulthood and older age groups. The majority were male (73.8%), and most participants resided in urban areas (74.9%). The ethnic distribution of participants was as follows: Janajati comprised the largest group (42.6%), followed by Brahmin/Chhetri (29.5%), Dalit (22.4%), and other ethnicities (5.5%). Regarding educational attainment, 45.9% of participants had completed secondary education or higher, 37.2% had up to lower secondary education, and 16.9% had no formal education. More than half of the participants (66.7%) were not engaged in any occupation. Most were married (65.6%) and belonged to families with four or fewer members (59.0%). Good knowledge about tuberculosis (TB) was reported by 79.8% of respondents, while 20.2% had poor knowledge. A history of contact with TB patients was reported by 26.8% of participants, and nearly one-third (31.1%) had a previous history of TB.

The prevalence of alcohol and tobacco use was 59.6% and 45.9%, respectively. Among the participants, 9.8% were diabetic, and 7.1% had hypertension. Based on BMI, 51.4% had normal weight, while 48.6% were categorized as underweight, overweight, or obese. When evaluating cumulative risk, 79.2% had at least one risk factor.

As reported in Table 2, there were significant differences between cases and controls with respect to education level, working status, family size, TB knowledge, known TB contact, previous TB history, and tobacco use (p < 0.05). All other variables including ethnicity, marital status, residence, alcohol use, diabetes, hypertension, and abnormal BMI were not significantly associated with MDR-TB (p ≥ 0.05).

The multivariate analysis presented in Table 2 identified significant associations between selected variables and the occurrence of MDR-TB. Individuals who were not currently working had significantly higher odds of having MDR-TB compared to those who were employed (AOR: 5.24, 95% CI: 1.33–20.64, p = 0.018). Furthermore, participants who reported known contact with a TB case were substantially more likely to have MDR-TB than those without such contact (AOR: 8.89, 95% CI: 2.46–32.15, p = 0.001). Other variables, including ethnicity, education level, marital status, family size, TB knowledge, previous TB history, and presence of cardiometabolic risk factors, did not show statistically significant associations in the final model (p ≥ 0.05).

**Table 1. Background characteristics of cases and controls.**

| Characteristics | Cases (N = 61) | Controls (N = 122) |
|---|---|---|
| | N (%) | N (%) |
| **Sex** | | |
| Female | 16 (26.23) | 32 (26.23) |
| Male | 45 (73.77) | 90 (73.77) |
| **Ethnicity** | | |
| Brahmin-Chhetri | 11 (18.03) | 43 (35.25) |
| Dalit | 14 (22.95) | 27 (22.13) |
| Janajati | 32 (52.46) | 46 (37.70) |
| Others | 4 (6.56) | 6 (4.92) |
| **Education** | | |
| No Education | 16 (26.23) | 15 (12.29) |
| Upto Lower Secondary | 24 (39.34) | 44 (36.07) |
| Secondary and above | 21 (34.43) | 63 (51.64) |
| **Working Status** | | |
| Yes | 7 (11.11) | 54 (44.26) |
| No | 54 (88.89) | 68 (55.74) |
| **Marital Status** | | |
| Married | 37 (60.66) | 83 (68.03) |
| Single | 24 (39.34) | 39 (31.97) |
| **Residence** | | |
| Rural | 16 (26.23) | 30 (24.59) |
| Urban | 45 (73.77) | 92 (75.41) |
| **Family Size** | | |
| ≤ 4 | 29 (47.54) | 79 (64.75) |
| > 4 | 32 (52.46) | 43 (35.25) |
| **TB Knowledge** | | |
| Good | 40 (65.57) | 106 (86.89) |
| Poor | 21 (34.43) | 16 (13.11) |
| **Contact with TB case** | | |
| Yes | 33 (54.10) | 16 (13.11) |
| No | 28 (45.90) | 106 (86.89) |
| **History of TB treatment** | | |
| Yes | 30 (49.18) | 27 (22.13) |
| No | 31 (50.82) | 95 (77.87) |
| **HIV** | | |
| Yes | 1 (1.64) | 1 (0.82) |
| No | 60 (98.36) | 121 (99.18) |
| **Alcohol Use** | | |
| Yes | 40 (65.57) | 69 (56.56) |
| No | 21 (34.43) | 53 (43.44) |
| **Tobacco Use** | | |
| Yes | 35 (57.38) | 49 (40.16) |
| No | 26 (42.62) | 73 (59.84) |
| **Diabetes** | | |
| Yes | 5 (8.20) | 13 (10.66) |
| No | 56 (91.80) | 109 (89.34) |

*(Continued)*

**Table 1.** (Continued)

| Characteristics | Cases (N=61) | Controls (N=122) |
| --- | --- | --- |
| | N (%) | N (%) |
| **Hypertension** | | |
| Yes | 5 (8.20) | 8 (6.56) |
| No | 56 (91.80) | 114 (93.44) |
| **BMI** | | |
| Abnormal | 33 (54.10) | 56 (45.90) |
| Normal | 28 (45.90) | 66 (54.10) |
| **Cumulative cardiometabolic risk** | | |
| At least one risk factor | 53 (86.89) | 92 (75.41) |
| No Risk factor | 8 (13.11) | 30 (24.59) |

## Discussion

Cardiometabolic comorbidities are rising rapidly in Nepal due to urbanization and various lifestyle factors, potentially adding to the TB epidemic. The occurrence of MDR-TB with such risk factors has important implications for disease progression, treatment outcomes, and health system planning. While most research has examined single risk factors, less is known about their combined effects like whether their coexistence amplifies susceptibility to MDR-TB or worsens prognosis. Thus, understanding these interactions is crucial for designing integrated TB–NCD programs, understand more about MDR-TB and targeted prevention as well as treatment strategies.

In this case-control study examining the co-occurrence of cardiometabolic and social risk factors in multidrug-resistant tuberculosis (MDR-TB), we found that cumulative cardiometabolic factors (diabetes, hypertension, abnormal BMI, tobacco use, and alcohol use) were not significantly associated with MDR-TB status. Instead, sociodemographic variables, particularly unemployment and a known contact with TB, emerged as strong predictors. This could be because resistance is primarily driven by treatment-related factors or transmission of resistant strains, with social and structural problems playing a greater role than individual metabolic risks in this setting.

While diabetes mellitus (DM) has been established as an independent risk factor for MDR-TB in several studies particularly for primary MDR-TB [18], but we did not find a significant association between DM and MDR-TB in our sample. This is consistent with other research suggesting no strong link between DM and drug resistance, particularly when assessing parameters like disease duration or glycemic control [19]. Similarly, although prior studies have linked tobacco smoking to MDR- and XDR-TB [20], and alcohol use has been associated with poor TB outcomes, these lifestyle factors may not exert a direct causal effect on resistance. Instead, interruptions in treatment due to alcohol use disorder, non-adherence, and other behavioral factors could indirectly contribute to resistance [9,21]. The evidence base on BMI and MDR-TB is also sparse; although low BMI is a known risk factor for adverse TB outcomes [22,23], its specific link to drug resistance remains inconclusive.

Most existing studies have focused on single risk factors or general TB risk rather than the cumulative effect of multiple cardiometabolic factors. A large Nepal-based study found that 43% of TB patients had at least one NCD risk factor, such as smoking, alcohol use, hypertension, obesity, or diabetes [24]. However, it did not assess drug resistance, and such aggregated risk scores may obscure opposing effects for example, DM increasing susceptibility while obesity might be neutral or protective. Biologically, cardiometabolic comorbidities do not directly cause drug resistance; resistance arises from pathogen mutation during inadequate treatment or transmission of resistant strains. It is thus plausible that cardiometabolic risks influence general TB vulnerability but add little specific risk for MDR-TB once TB is acquired. Methodological factors such as sample size and variable aggregation may also contribute to null findings.

**Table 2. Multivariable logistic regression analysis of risk factors associated with MDR-TB.**

| Characteristics | COR (CI) | P value | AOR (CI) | P value |
|---|---|---|---|---|
| **Ethnicity** | | | | |
| Brahmin-Chhetri | Ref | | | |
| Dalit | 1.99 (0.77-5.12) | 0.153 | 0.53 (0.13-2.24) | 0.388 |
| Janajati | 2.82 (1.22-6.50) | 0.015 | 1.28 (0.36-4.49) | 0.705 |
| Others | 2.76 (0.66-11.50) | 0.162 | 1.47 (0.16-13.25) | 0.731 |
| **Education** | | | | |
| Secondary and above | Ref | | | |
| Upto Lower Secondary | 2.48 (1.08-5.67) | **0.032**** | 5.26 (0.78-35.52) | 0.089 |
| No Education | 6.72 (2.09-21.62) | **0.001**** | 1.52 (0.42-5.54) | 0.526 |
| **Working Status** | | | | |
| Yes | Ref | | | |
| No | 7.00 (2.67-18.32) | **<0.001*** | 5.24 (1.33-20.64) | **0.018**** |
| **Marital Status** | | | | |
| Married | Ref | | | |
| Single | 1.84 (0.75-4.49) | 0.181 | 3.30 (0.74-14.74) | 0.117 |
| **Residence** | | | | |
| Urban | Ref | | | |
| Rural | 1.11 (0.51-2.39) | 0.793 | – | – |
| **Family Size** | | | | |
| ≤ 4 | Ref | | | |
| > 4 | 2.14 (1.10-4.15) | **0.024**** | 2.75 (0.91-8.35) | 0.074 |
| **TB Knowledge** | | | | |
| Good | Ref | | | |
| Poor | 3.23 (1.54-6.79) | **0.002**** | 2.30 (0.63-8.42) | 0.207 |
| **Contact with TB case** | | | | |
| No | Ref | | | |
| Yes | 8.64 (3.58-20.85) | **<0.001*** | 8.89 (2.46-32.15) | **0.001*** |
| **History of TB treatment** | | | | |
| No | Ref | | | |
| Yes | 3.47 (1.73-6.95) | **<0.001*** | 2.33 (0.78-7.00) | 0.132 |
| **HIV** | | | | |
| No | Ref | | | |
| Yes | 1.00 (0.09-11.03) | 1.000 | – | – |
| **Alcohol Use** | | | | |
| No | Ref | | | |
| Yes | 2.13 (0.86-5.30) | 0.103 | – | – |
| **Tobacco Use** | | | | |
| No | Ref | | | |
| Yes | 2.74 (1.23-6.08) | **0.013**** | – | – |
| **Diabetes** | | | | |
| No | Ref | | | |
| Yes | 0.73 (0.24-2.24) | 0.585 | – | – |
| **Hypertension** | | | | |
| No | Ref | | | |
| Yes | 1.31 (0.38-4.45) | 0.671 | – | – |

*(Continued)*

**Table 2.** (Continued)

| Characteristics | COR (CI) | P value | AOR (CI) | P value |
|---|---|---|---|---|
| **BMI** | | | | |
| Normal | Ref | | | |
| Abnormal | 1.44 (0.75-2.74) | 0.271 | – | – |
| **Cumulative cardiometabolic risk** | | | | |
| No Risk factor | Ref | | | |
| At least one risk factor | 3.04 (1.12-8.28) | **0.029**** | 2.76 (0.54-14.17) | 0.223 |

COR: Crude Odds Ratio; CI: Confidence Interval; AOR: Adjusted Odds Ratio.

* Indicates statistical significance (p<0.001).

** Indicates statistical significance (p<0.05).

In contrast, our finding that social determinants—unemployment and TB contact—predict MDR-TB aligns with global literature. Unemployment has been repeatedly associated with TB and MDR-TB risk, likely due to its relationship with poverty, overcrowded living conditions, poor nutrition, and reduced healthcare access. Daily wage laborers and unemployed individuals have been shown to carry a disproportionate burden of MDR-TB [25]. Likewise, known TB contact emerged as a powerful predictor in our study, underscoring the central role of transmission in resistance dynamics. Previous research shows that individuals exposed to MDR-TB cases are dramatically more likely to develop MDR-TB themselves [26,27], with adjusted odds ratios as high as 75.2 [27]. These data emphasize that MDR-TB is often not the result of acquired resistance via metabolic vulnerability but rather reflects ongoing transmission chains.

Taken together, our findings carry several public health implications. While screening for NCDs remains essential in integrated TB care, such screening may have limited predictive value for MDR-TB specifically. Instead, interventions should emphasize contact tracing, prompt diagnosis, and social protection measures. Addressing structural determinants such as unemployment, inadequate housing, and food insecurity may yield more substantial gains in MDR-TB prevention. Programs aimed at identifying and supporting individuals with known TB contact, particularly in households or communities affected by resistant strains, should be prioritized to break transmission chains.

The sample size was relatively small, which may have limited the power to detect statistically significant associations for some variables. Being a case-control study, it is also prone to recall bias, particularly regarding previous TB episodes, treatment history, and contact with TB cases. Although we examined five cardiometabolic risk factors (hypertension, diabetes, smoking, alcohol use, and abnormal BMI) in a cumulative manner, these factors may be interrelated (e.g., obesity and diabetes), potentially confounding individual effects. Future research should explore these dynamics with larger, longitudinal studies that can assess whether cardiometabolic factors influence treatment failure or relapse and pathways through which resistance could still emerge. Research into how the immune system is affected in people with diabetes or metabolic syndrome may help explain the biological pathways involved.

## Conclusion

This study highlights the complex interplay of sociodemographic, behavioral, and metabolic factors in the development of MDR-TB. While unemployment and contact to TB case were significantly associated with MDR-TB, cumulative cardiometabolic risk factors did not show an independent association in multivariate analysis. Further research, particularly longitudinal studies with larger sample sizes, is needed to better understand the role of metabolic factors and their interactions with other determinants in the pathogenesis of MDR-TB. Given the high burden of MDR-TB in Nepal, these findings underscore the need for integrated TB and NCD care models that address both traditional and emerging risk factors.

## Acknowledgments

We express our gratitude to NHRC, Dr. Abhiskar Gautam and Tuberculosis Treatment Center staffs for their support. We appreciate the time contributed by participants for this important study.

## Author contributions

**Conceptualization:** Sishir Poudel.

**Data curation:** Sishir Poudel, Laxman Wagle, Tara Prasad Aryal.

**Formal analysis:** Sishir Poudel, Laxman Wagle, Binay Adhikari, Sushan Pokharel, Dipendra Adhikari, Kshitiz Bhandari, Kshitiz Rijal, Jyoti Bastola Paudel.

**Investigation:** Sishir Poudel, Tara Prasad Aryal.

**Methodology:** Sishir Poudel, Laxman Wagle, Tara Prasad Aryal, Binay Adhikari, Sushan Pokharel, Dipendra Adhikari, Kshitiz Bhandari, Kshitiz Rijal, Jyoti Bastola Paudel.

**Project administration:** Laxman Wagle.

**Resources:** Sishir Poudel.

**Software:** Sishir Poudel.

**Supervision:** Sishir Poudel, Laxman Wagle, Tara Prasad Aryal, Binay Adhikari, Sushan Pokharel, Dipendra Adhikari, Kshitiz Bhandari, Kshitiz Rijal, Jyoti Bastola Paudel.

**Validation:** Sishir Poudel, Tara Prasad Aryal.

**Visualization:** Sishir Poudel, Laxman Wagle, Binay Adhikari, Sushan Pokharel.

**Writing – original draft:** Sishir Poudel.

**Writing – review & editing:** Sishir Poudel, Laxman Wagle, Tara Prasad Aryal, Binay Adhikari, Sushan Pokharel, Dipendra Adhikari, Kshitiz Bhandari, Kshitiz Rijal, Jyoti Bastola Paudel.

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
