## [Decision Letter · Decision Letter 0]

21 Jul 2025

PONE-D-25-29263Cardiometabolic risk factors associated with multidrug resistant tuberculosis: A case-control studyPLOS ONE

Dear Dr. Poudel,

Thank you for submitting your manuscript to PLOS ONE. After careful consideration, we feel that it has merit but does not fully meet PLOS ONE’s publication criteria as it currently stands. Therefore, we invite you to submit a revised version of the manuscript that addresses the points raised during the review process.

**ACADEMIC EDITOR: ** Please address all the comments raised by the reviewers to improve the qaualiy of your manuscript, mainly rasied the methodology to select cases and controls and the result and discusion also should be enriched.

We look forward to receiving your revised manuscript.

Kind regards,

Tsegaye Alemayehu, Msc

Academic Editor

PLOS ONE

Journal Requirements:

1. Please ensure that your manuscript meets PLOS ONE's style requirements, including those for file naming. The PLOS ONE style templates can be found at https://journals.plos.org/plosone/s/file?id=wjVg/PLOSOne_formatting_sample_main_body.pdf and https://journals.plos.org/plosone/s/file?id=ba62/PLOSOne_formatting_sample_title_authors_affiliations.pdf.

Reviewers' comments:

Reviewer's Responses to Questions

**Comments to the Author**

1. Is the manuscript technically sound, and do the data support the conclusions?

Reviewer #1: Yes

Reviewer #2: Partly

2. Has the statistical analysis been performed appropriately and rigorously? 

Reviewer #1: Yes

Reviewer #2: No

3. Have the authors made all data underlying the findings in their manuscript fully available?

Reviewer #1: Yes

Reviewer #2: Yes

4. Is the manuscript presented in an intelligible fashion and written in standard English?

Reviewer #1: Yes

Reviewer #2: Yes

5. Review Comments to the Author

Reviewer #1: I have no major concerns regarding the study or the manuscript. The idea and objectives of the study are clear and well-defined, and the results are presented effectively. The discussion section is appropriate.

However, I have a few minor suggestions: I would appreciate some contextual information on the TB situation in Nepal, specifically concerning the incidence, prevalence, and percentage of multidrug-resistant TB cases. Additionally, I noticed a lack of data regarding substance use among the study participants. This is a crucial risk factor for both TB and drug-resistant TB. Is there no substance use problem in Nepal?

Reviewer #2: Review report

Title: Cardiometabolic Risk Factors Associated with Multidrug-Resistant Tuberculosis: A Case-Control Study

General Comments

This case-control study aims to identify factors associated with multidrug-resistant tuberculosis (MDR-TB). Cases (patients with MDR-TB) and controls (patients with drug-sensitive TB, DS-TB) were included, and data were collected using a pre-tested questionnaire. The questionnaire covered a range of exposure factors: sociodemographic, cardiometabolic (alcohol, tobacco use, hypertension, diabetes, abnormal body mass index [BMI]), and individual factors (previous TB exposure, known contact with a TB case, knowledge about TB).

The topic is relevant, as it addresses the dual burden of infectious and non-communicable diseases among TB patients. Identifying individual risk factors associated with MDR-TB may contribute to more targeted and effective prevention strategies.

However, several concerns need to be addressed:

- The method used to select cases and controls should be clearly described.

- The statistical analysis section requires more detailed explanations to improve the interpretability of the results.

- Additional subgroup analyses (based on age, sex, residence area), depending on sample size, could strengthen the findings.

Minor Comments

1. Abstract

Avoid using abbreviations in the abstract. Terms such as multidrug-resistant tuberculosis (MDR-TB) and drug-sensitive tuberculosis (DS-TB) should be written in full.

Line 29: Replace BMI with overweight/obesity or abnormal BMI.

2. Introduction

Include epidemiological data on MDR-TB in the country where the study was conducted.

Additional arguments could strengthen the rationale for the study, such as the co-occurrence of cardiometabolic risk factors with infections like HIV, parasitic diseases, or viral hepatitis.

3. Materials and Methods

Line 139: Explain how blood pressure was measured and specify the classification criteria used to define hypertension.

Lines 141–142: Mention the reference used for the classification of fasting blood glucose levels.

Clearly describe how cases and controls were selected.

In the statistical analysis section, please:

- Specify the statistical tests used (e.g., chi-square, t-test, logistic regression).

- Define the dependent and independent variables.

Indicate how variables were included in the multivariate model (e.g., manual, automatic, stepwise).

Justify the use of multivariate analysis when the bivariate analysis does not show significant associations between cardiometabolic risk factors and MDR-TB.

4. Results

Line 192: Add the ± symbol before the standard deviation value.

Line 207 (Table 1): Check that the percentages for each variable add up to 100%. For example, under “ethnic group,” the sum is incorrect (17.46% + 22.22% + 50.79% + 6.63% = 96.82%).

Lines 207 and 222 (Tables 1 and 2): Use a consistent scientific table format with three lines (header, subtitle, total).

Table 2: Present both crude and adjusted Odds Ratios to show how associations evolve after adjustment.

Ensure consistent formatting of p-values, using italicized p.

The results section is relatively brief and could be strengthened by subgroup analyses by age, sex, or area of residence.

5. Discussion

Begin this section by contextualizing the topic and highlighting the public health implications of MDR-TB comorbidities with cardiometabolic risk factors.

Discuss the combined effects of specific risk factors (e.g., those defining metabolic syndrome) in relation to your findings.

Clearly state the limitations of the study (e.g., selection bias, recall bias, small sample size, self-reported data) at the end of this section.

Major Comments

Title: Consider rephrasing the title for clarity. Suggested title: Association Between Cardiometabolic Risk Factors and Multidrug-Resistant Tuberculosis: A Case-Control Study.

Methodology:

Clearly describe how cases and controls were selected.

Provide details of the statistical tests used.

Clearly define the model used for multivariate analysis, including dependent and independent variables, and the method for including covariates.

Multivariate Analysis: If the bivariate analysis does not show a significant association between cumulative cardiometabolic risk factors and MDR-TB, the rationale for conducting multivariate analysis should be explicitly stated—based, for example, on clinical relevance or prior literature.

Final Recommendation

This manuscript addresses a relevant and timely topic at the intersection of infectious and non-communicable diseases. The study's aim—to explore the association between cardiometabolic risk factors and multidrug-resistant tuberculosis (MDR-TB)—is commendable and could provide valuable insights for targeted prevention strategies. However, the current version of the manuscript requires substantial revisions before it can be considered for publication.

The main concerns relate to the lack of clarity in the methodology, particularly in the selection of cases and controls, and the insufficient detail in the statistical analysis. The presentation of results could be improved with subgroup analyses and better-structured tables. The discussion section would also benefit from a more in-depth exploration of the public health implications and clearer articulation of study limitations.

Therefore, my recommendation is: Major Revision Required

The authors are encouraged to address both major and minor concerns outlined in this review and to resubmit a revised version that enhances methodological transparency, strengthens the analysis, and improves the overall scientific rigor of the manuscript.

6. PLOS authors have the option to publish the peer review history of their article (what does this mean? ). If published, this will include your full peer review and any attached files.

**Do you want your identity to be public for this peer review?** For information about this choice, including consent withdrawal, please see our Privacy Policy .

Reviewer #1: No

Reviewer #2: **Yes: ** Cyrille Bisseye, PhD

---

## [Author Response · Author response to Decision Letter 1]

24 Aug 2025

Response to the academic editor

Our responses are given in line with the comment numbers mentioned earlier.

1. We have formatted the manuscript according to PLOS ONE’s style requirement.

2. We have uploaded our data to figshare. Link: http://dx.doi.org/10.6084/m9.figshare.28931690.v1

3. N/A

Response to reviewers

Reviewer #1: I have no major concerns regarding the study or the manuscript. The idea and objectives of the study are clear and well-defined, and the results are presented effectively. The discussion section is appropriate.

However, I have a few minor suggestions: I would appreciate some contextual information on the TB situation in Nepal, specifically concerning the incidence, prevalence, and percentage of multidrug-resistant TB cases. Additionally, I noticed a lack of data regarding substance use among the study participants. This is a crucial risk factor for both TB and drug-resistant TB. Is there no substance use problem in Nepal?

Response: Thank you for your comments. We have added the contextual information in the introduction section. In our study, we focused mainly on alcohol and smoking as the primary forms of substance use. While we acknowledge that other forms of substance use exist in Nepal, their prevalence among our study participants was very low, and we were unable to establish a meaningful association with tuberculosis.”.

Reviewer #2: Review report

Title: Cardiometabolic Risk Factors Associated with Multidrug-Resistant Tuberculosis: A Case-Control Study

General Comments

This case-control study aims to identify factors associated with multidrug-resistant tuberculosis (MDR-TB). Cases (patients with MDR-TB) and controls (patients with drug-sensitive TB, DS-TB) were included, and data were collected using a pre-tested questionnaire. The questionnaire covered a range of exposure factors: sociodemographic, cardiometabolic (alcohol, tobacco use, hypertension, diabetes, abnormal body mass index [BMI]), and individual factors (previous TB exposure, known contact with a TB case, knowledge about TB).

The topic is relevant, as it addresses the dual burden of infectious and non-communicable diseases among TB patients. Identifying individual risk factors associated with MDR-TB may contribute to more targeted and effective prevention strategies.

However, several concerns need to be addressed:

- The method used to select cases and controls should be clearly described.

- The statistical analysis section requires more detailed explanations to improve the interpretability of the results.

- Additional subgroup analyses (based on age, sex, residence area), depending on sample size, could strengthen the findings.

Minor Comments

1. Abstract

Avoid using abbreviations in the abstract. Terms such as multidrug-resistant tuberculosis (MDR-TB) and drug-sensitive tuberculosis (DS-TB) should be written in full.

Line 29: Replace BMI with overweight/obesity or abnormal BMI.

Response: Thank you for the comment. We have written the respective terms in full. We also replaced BMI with abnormal BMI.

2. Introduction

Include epidemiological data on MDR-TB in the country where the study was conducted.

Response: Thank you for the comment. We have added the epidemiological data on MDR-TB.

Additional arguments could strengthen the rationale for the study, such as the co-occurrence of cardiometabolic risk factors with infections like HIV, parasitic diseases, or viral hepatitis.

Response: Thank you for the comment. We searched the literature but couldn’t find the relevant studies.

3. Materials and Methods

Line 139: Explain how blood pressure was measured and specify the classification criteria used to define hypertension.

Response: Thank you for the comment. We have added the method and criteria in the specific section of methodology.

Lines 141–142: Mention the reference used for the classification of fasting blood glucose levels.

Response: Thank you for the comment. We have added the reference in the specific section of methodology.

Clearly describe how cases and controls were selected.

Response: Thank you for the comment. We have described the selection of cases and controls in the specific section of methodology.

In the statistical analysis section, please:

- Specify the statistical tests used (e.g., chi-square, t-test, logistic regression).

Response: Thank you for the comment. We have included a detailed description in data analysis section.

- Define the dependent and independent variables.

Response: Thank you for the comment. We have modified the methodology section defining the dependent and independent variables.

Indicate how variables were included in the multivariate model (e.g., manual, automatic, stepwise).

Response: Thank you for the comment. We have explained in detail about the selection of variables for multivariate analysis in the data analysis section.

Justify the use of multivariate analysis when the bivariate analysis does not show significant associations between cardiometabolic risk factors and MDR-TB.

Response: Thank you for the comment. We apologize for a minor error in preparing the manuscript and tables. The bivariate analysis of cardiometabolic risk factors and MDR-TB was incorrectly reported as not significant; in fact, the analysis showed a significant association. We have corrected this in the revised tables and text.

4. Results

Line 192: Add the ± symbol before the standard deviation value.

Response: Thank you for the comment. We have added the symbol.

Line 207 (Table 1): Check that the percentages for each variable add up to 100%. For example, under “ethnic group,” the sum is incorrect (17.46% + 22.22% + 50.79% + 6.63% = 96.82%).

Response: Thank you for the comment. We apologize for the error. It was incorrectly calculated taking sample size as 63. We have corrected the error.

Lines 207 and 222 (Tables 1 and 2): Use a consistent scientific table format with three lines (header, subtitle, total).

Table 2: Present both crude and adjusted Odds Ratios to show how associations evolve after adjustment.

Ensure consistent formatting of p-values, using italicized p.

Response: Thank you for the comments. We have modified the table according to PLOS ONE guidelines. We have revised the table showing both crude and adjusted odds ratio using appropriate formatting for the p-values.

The results section is relatively brief and could be strengthened by subgroup analyses by age, sex, or area of residence.

Response: Thank you for the comment. We conducted subgroup analyses; however, age and sex were not included as subgroups because these variables were matched in the study design. Additionally, the small overall sample size and limited numbers within subgroups limited our findings.

5. Discussion

Begin this section by contextualizing the topic and highlighting the public health implications of MDR-TB comorbidities with cardiometabolic risk factors.

Discuss the combined effects of specific risk factors (e.g., those defining metabolic syndrome) in relation to your findings.

Response: Thank you for the comments. We have modified the discussion section starting with the context and, we have discussed the effects of the risk factors.

Clearly state the limitations of the study (e.g., selection bias, recall bias, small sample size, self-reported data) at the end of this section.

Response: Thank you for the comment. We have added the limitation at the end of discussion section.

Major Comments

Title: Consider rephrasing the title for clarity. Suggested title: Association Between Cardiometabolic Risk Factors and Multidrug-Resistant Tuberculosis: A Case-Control Study.

Response: Thank you for the comment. We agree with your title suggestion and have modified it for clarity.

Methodology:

Clearly describe how cases and controls were selected.

Provide details of the statistical tests used.

Clearly define the model used for multivariate analysis, including dependent and independent variables, and the method for including covariates.

Multivariate Analysis: If the bivariate analysis does not show a significant association between cumulative cardiometabolic risk factors and MDR-TB, the rationale for conducting multivariate analysis should be explicitly stated—based, for example, on clinical relevance or prior literature.

Response: Thank you for the comments. These comments have been addressed in the relevant section of methodology.

Final Recommendation

This manuscript addresses a relevant and timely topic at the intersection of infectious and non-communicable diseases. The study's aim—to explore the association between cardiometabolic risk factors and multidrug-resistant tuberculosis (MDR-TB)—is commendable and could provide valuable insights for targeted prevention strategies. However, the current version of the manuscript requires substantial revisions before it can be considered for publication.

The main concerns relate to the lack of clarity in the methodology, particularly in the selection of cases and controls, and the insufficient detail in the statistical analysis. The presentation of results could be improved with subgroup analyses and better-structured tables. The discussion section would also benefit from a more in-depth exploration of the public health implications and clearer articulation of study limitations.

Therefore, my recommendation is: Major Revision Required

The authors are encouraged to address both major and minor concerns outlined in this review and to resubmit a revised version that enhances methodological transparency, strengthens the analysis, and improves the overall scientific rigor of the manuscript, the manuscript would be suitable for publication in its current journal.

Response: We sincerely thank you for your thoughtful comments and suggestions. We have revised the manuscript accordingly, and we believe that your valuable feedback has significantly improved the quality of our work. The review process has also been a great learning experience for us as authors.

Other edits

We have edited the contents on “Abstract” section and some minor corrections in the manuscript content.

---

## [Decision Letter · Decision Letter 1]

14 Sep 2025

Association between cardiometabolic risk factors and multidrug-resistant tuberculosis: A case-control study

PONE-D-25-29263R1

Dear Dr. Poudel,

We’re pleased to inform you that your manuscript has been judged scientifically suitable for publication and will be formally accepted for publication once it meets all outstanding technical requirements.

Kind regards,

Tsegaye Alemayehu, Msc

Academic Editor

PLOS ONE

Additional Editor Comments (optional):

Reviewer #1:

Reviewer #2:

Reviewers' comments:

Reviewer's Responses to Questions

**Comments to the Author**

1. If the authors have adequately addressed your comments raised in a previous round of review and you feel that this manuscript is now acceptable for publication, you may indicate that here to bypass the “Comments to the Author” section, enter your conflict of interest statement in the “Confidential to Editor” section, and submit your "Accept" recommendation.

Reviewer #1: All comments have been addressed

Reviewer #2: All comments have been addressed

2. Is the manuscript technically sound, and do the data support the conclusions?

Reviewer #1: (No Response)

Reviewer #2: Yes

3. Has the statistical analysis been performed appropriately and rigorously? 

Reviewer #1: (No Response)

Reviewer #2: (No Response)

4. Have the authors made all data underlying the findings in their manuscript fully available?

Reviewer #1: (No Response)

Reviewer #2: Yes

5. Is the manuscript presented in an intelligible fashion and written in standard English?

Reviewer #1: (No Response)

Reviewer #2: Yes

6. Review Comments to the Author

Reviewer #1: (No Response)

Reviewer #2: To

The Editor

PLOS ONE

Subject: Final Recommendation for Manuscript Entitled “Association between cardiometabolic risk factors and multidrug-resistant tuberculosis: A case-control study”

Dear Editor,

I am writing regarding the revised manuscript entitled “Association between cardiometabolic risk factors and multidrug-resistant tuberculosis: A case-control study.”

The authors have carefully and satisfactorily addressed all the comments and suggestions raised during the peer-review process. The revisions have strengthened the manuscript, improved clarity, and ensured that the methodology, results, and interpretations are clearly and appropriately presented.

Based on the comprehensive responses provided by the authors and the quality of the current version, In my point of view, I confirm that the manuscript is now suitable for publication.

Thank you for your attention.

Sincerely,

Cyrille BISSEYE

Associate Professor, Head of the Laboratory of Molecular and Cellular Biology

University of Science and Technology of Masuku, Franceville, Gabon

7. PLOS authors have the option to publish the peer review history of their article (what does this mean? ). If published, this will include your full peer review and any attached files.

**Do you want your identity to be public for this peer review?** For information about this choice, including consent withdrawal, please see our Privacy Policy .

Reviewer #1: No

Reviewer #2: **Yes: ** Associate Prof Cyrille BISSEYE

---

## [Editor Report · Acceptance letter]

PONE-D-25-29263R1

PLOS ONE

Dear Dr. Poudel,

I'm pleased to inform you that your manuscript has been deemed suitable for publication in PLOS ONE. Congratulations! Your manuscript is now being handed over to our production team.

Kind regards,

on behalf of

Dr. Tsegaye Alemayehu

Academic Editor

PLOS ONE